# Enhancing Small Medical Dataset Classification Performance Using GAN

Mohammad Alauthman [1] , Ahmad Al-qerem [2] , Bilal Sowan [3] , Ayoub Alsarhan [4] , Mohammed Eshtay [5], Amjad Aldweesh [6,*] and Nauman Aslam [7]

1   Department of Information Security, Faculty of Information Technology, University of Petra, Amman 11196, Jordan
2   Computer Science Department, Faculty of Information Technology, Zarqa University, Zarqa 13110, Jordan
3   Department of Business Intelligence and Data Analytics, University of Petra, Amman 11196, Jordan
4   Department of Information Technology, Faculty of Prince Al-Hussein Bin Abdallah II for Information Technology, The Hashemite University, Zarqa 13133, Jordan
5   Abdul Aziz Al Ghurair School of Advanced Computing (ASAC), Luminus Technical University, Amman 11118, Jordan
6   College of Computing and Information Technology, Shaqra University, Riyadh 11911, Saudi Arabia
7   Department of Computer Science and Digital Technologies, Faculty of Engineering and Environment, Northumbria University, Newcastle upon Tyne NE1 8ST, UK
*   Correspondence: a.aldweesh@su.edu.sa

**Abstract:** Developing an effective classification model in the medical field is challenging due to limited datasets. To address this issue, this study proposes using a generative adversarial network (GAN) as a data-augmentation technique. The research aims to enhance the classifier's generalization performance, stability, and precision through the generation of synthetic data that closely resemble real data. We employed feature selection and applied five classification algorithms to thirteen benchmark medical datasets, augmented using the least-square GAN (LS-GAN). Evaluation of the generated samples using different ratios of augmented data showed that the support vector machine model outperforms other methods with larger samples. The proposed data augmentation approach using a GAN presents a promising solution for enhancing the performance of classification models in the healthcare field.

**Keywords:** data augmentation; GANs; medical dataset; machine learning; healthcare

## 1. Introduction

Computational intelligence or computer intelligence is being used more frequently in medical diagnosis. Machine intelligence-assisted decision-making systems are often used to help doctors diagnose a patient's illness, but they do not replace doctors. Most of a doctor's knowledge comes from the symptoms and diagnoses of her patients. Thus, diagnostic accuracy is highly dependent on the doctor's experience. Computerized medical decision support systems help clinicians make quick, accurate diagnoses [1,2].

In machine learning, data augmentation is recurrently used to generate fake training data when only small training sets are available, especially in object recognition, handwriting recognition, and speech recognition [3–6].

Errors in the classifiers caused by variance can be mitigated by enhancing the training data. Data augmentation can help to reduce model overfitting caused by a lack of learning examples. Many different types of generative adversarial networks (GANs) have been proposed. Although numerous examples of GAN-based data-augmentation techniques are being used in the literature, the majority of work has been performed on images, handwriting, and speech. Data augmentation of tabular data for classification purposes has not yet been studied much [7,8].

This research used a GAN-based data augmentation framework to show how different classification models work with small tabular data. Least-square GAN (LS-GAN) creates fake training data so that a large amount of field data is no longer needed.

The motivation for this work was to address the problem of limited training data in the medical field, which makes it challenging to develop an effective classification model. The study's objective was to utilize a GAN as a data-augmentation technique to improve the classifier's generalization performance, stabilize the classifier, and provide a more precise indication of the number of samples needed by the model, thereby avoiding overfitting due to the small number of the labeled samples. The study aimed to highlight the potential of GANs for enhancing the performance of classification models in the healthcare field by presenting a new approach for data augmentation in small medical datasets.

There are many areas of research that focus on making classification models more general. The generalizability of a model is measured by comparing its performance on training data with that on new data (testing data). Weakly generalizable models overfit data in the training set. Overfitting can be discovered by checking the accuracy on the training and validation data at various times in the training process [9–11].

Most previous works have concentrated on a small (original) dataset. However, obtaining such a dataset in the medical field is challenging due to annotated data shortage. Thus, this research proposes a deep learning-based framework to generate more real data from existing data with a generative adversarial network (GAN). This paper focuses on improving the generalizability of the models built using small medical datasets. The contribution summary of this paper is as follows.

1. We proposed a general methodology for small medical data classification that deploys an augmentation technique and a feature selection strategy, followed by appropriate training of a suitable classifier algorithm.
2. Brouta feature selection is utilized as a feature selection strategy with multiple machine learning algorithms.
3. We presented highly accurate classifiers for 13 different diseases and suggested a generalized augmentation approach that should perform well for other similar datasets.

The structure of the paper is as follows: Section 2 provides an overview of the state-of-the-art studies in medical data categorization and data augmentation, serving as context for the research and indicating research gaps. Section 3 outlines the methodology employed, describing the primary components of the proposed framework. Section 4 presents the findings, and Section 5 offers a summary of the results and main contribution of the work.

## 2. Related Works

In recent years, data augmentation has been widely studied as an effective strategy to improve the performances of machine learning algorithms [12,13]. Data augmentation is a technique that generates new, synthetic samples by applying various transformations to the original data. This approach has been particularly popular in the field of computer vision, where image transformations such as rotation, flipping, and cropping are commonly used to increase the number and diversity of training samples [7]. In the medical field, data augmentation is also crucial because of the limited availability of data [14,15]. In this paper, we review the literature on data-augmentation techniques that have been proposed for medical data classification and analyze their effectiveness at improving the performances of machine learning algorithms.

Abbass et al. [16] suggested a method for diagnosing breast cancer with the Pareto-differential evaluation algorithm with a local search scheme called the memetic Pareto artificial neural network (MPANN). Following that, Kiyan et al. [17] presented a computational neural network-based approach to breast cancer diagnosis. Karabatak et al. [18] developed an expert method for breast cancer diagnosis, where association rules (ARs) were used to reduce the dataset dimensions.

Peng et al. [19] proposed a hybrid feature-selection method to address the high-dimensionality issues of biomedical data and tested it with a breast cancer dataset. Fana

et al. [20] combined case-based data clustering with a blurred decision tree to create a hybrid model for medical data classification. Two datasets of WBC and liver disorders, respectively, were performed on the platform.

Azar et al. [21] introduced three methods of classification, namely, the radial base function (RBF), multilayer perceptron (MLP), and probabilistic neural network (PNN), and experimented with a dataset of breast cancer. Moreover, the PNN showed better performance in their studies than the MLP. Numerous studies on medical data classification have been published in the literature over the past three years, but only on a breast cancer dataset.

Cubuk et al. [22] suggested AutoAugment, which uses reinforcement learning (RL) to identify a data augmentation policy when a target dataset and a model are provided. A RNN is used to store augmentation policies, and the model is trained using the policies. After training, an assessment for augmentation policies is given as a reward for updating the controller. AutoAugment improved the performances of numerous picture recognition benchmarks dramatically. However, even in a simplified situation, where the target dataset and the network size are minimal, the workload takes enormous amounts of GPU time.

In one paper [23], the data augmentation method was used to train a convolutional neural network (CNN). This approach's key challenge is the occlusion problem by enhancing CNN generalization. In the training process, the algorithm chooses a random region size of pixels from a random image and replaces it with random pixel values.

In the research of Xie et al. [24], a data augmentation strategy was used in the semi-supervised learning environment for unlabeled data. The process of using unsupervised methods to improve data is known as unsupervised data augmentation (UDA). When training with real unlabeled data, UDA improves model consistency while learning real unlabeled data. UDA employs actual noise generated by methods similar to earlier data augmentation methods, and it takes steps to reduce the KL divergence between real and augmented data.

In [25], he authors utilized a new Bayesian data augmentation strategy. The new data are assumed to be random, annotated data points. This strategy enhanced the results by training the generator distribution using the training and classification models.

Lim et al. [26] proposed the fast AutoAugment method. The fast AutoAugment method was suggested, which takes inspiration from Bayesian data augmentation. To deal with enhanced data, fast AutoAugment considers the augmented data as if they were absent, and these points are restored by using a family of inference-time augmentations. The Bayesian method has been applied to help speed up this search. Many datasets were examined, and it was shown that the search time was quicker than that of AutoAugment, and there was a negligible increase in error rate.

Population-based augmentation (PBA) is a method proposed by Daniel et al. [27] to replace the previous ones and present an augmentation policy. PBA generated the optimal augmentation policy schedule for each training period. Using a PBA algorithm is advantageous, since no intensive processing resources are required. Based on experiments, it has been shown that PBA-specific training takes less time than training using other methods.

The study [28] presented a new conditional generative adversarial network (cGAN) to create models with the ability to generate images under certain conditions. Additionally, compared to a regular GAN, the prediction and accuracy are improved. The proper selection of the loss function automatically improves the system's efficiency. The suggested model outperforms state-of-the-art approaches by a margin of 74% and only needs around 10 min to be trained, even on a large dataset.

One interesting approach is a random erasing technique proposed in [29]. This method is fast and easy to use, and it helps CNNs learn to generalize. In this method, a noise-filled rectangle is painted randomly on an image, which changes the values of the pixels. As the authors indicated, adding occlusion photos to the dataset decreases overfitting and

strengthens the model. Table 1 summarizes some of the research approaches that apply augmentation techniques.

**Table 1.** Data augmentation-related work.

| No. | Research | Datasets | Method and Results |
|-----|----------|----------|--------------------|
| 1 | Daniel et al. [27]: Population Augmentation policy | SVHN | Regression and the error: 0.1 |
| 2 | Zhong et al. [29]: Augmentation by Data Random Erasing | CIFAR-10 | Regression and the error: 0.31 |
| 3 | Lim et al. [26]: Fast AutoAugment. | CIFAR-10 | Regression and the error: 0.20 |
| 4 | Cubuk et al. [22]: AutoAugment data strategies. | SVHN and ImageNet | Classification accuracy rates: 0.835 Regression and the error: 0.10% |
| 5 | Xie et al. [24]: Unsupervised Data Augmentation. | CIFAR-10 | Classification accuracy rates: 0.79 Regression and the error rate: 0.3 |
| 6 | Tran et al. [25]:A Bayesian Data Augmentation. | CIFAR-10 | Classification and Accuracy: 0.93 |

In conclusion, the literature review on data augmentation for medical data classification shows that various techniques have been proposed and evaluated in recent years; however, most of the research has been focused on large datasets and not specifically on small medical datasets. There is a gap in the current research on how to effectively use data-augmentation techniques on small medical datasets, where the scarcity of data is one of the main challenges. Additionally, while some studies have reported improved performance with data augmentation, there is a lack of understanding of the optimal number of augmented samples for small medical datasets, and more research is needed to determine the optimal balance between the number of augmented samples and the risk of overfitting. Additionally, it is not well understood how the different data-augmentation techniques compare and perform when applied to small medical datasets. This is an important area for future research, as small medical datasets are becoming increasingly more common.

## 3. Materials and Methods

### 3.1. Datasets

In this study, thirteen medical classification datasets were utilized to evaluate the performances of various algorithms. The datasets were obtained from the UCI Machine Learning repository [30] and Keel [31], and varied in terms of the number of features and instances. The datasets were chosen to test the proposed strategies at different levels of complexity. Some datasets, such as Spectf, have a large number of features; and others, such as Blood, have fewer features. Additionally, the number of instances also varies. Some datasets, such as Parkinson's, have a limited number of cases; and others, such as Phoneme, have larger numbers of instances. These datasets were used to test how accurate the algorithms are with varying augmentation techniques and to validate the proposed GAN-based data augmentation framework. Table 2 provides a summary of the datasets. These datasets represent a diverse range of medical classification problems and allow for testing the effectiveness of the proposed strategies on different types of data.

### 3.2. Model Construction Overview

Figure 1 depicts the overall architecture of the designed system for data augmentation and classifications. This methodology first separates raw data into training and test data. For data augmentation, the raw training data are utilized. The machine learning algorithms are trained using a combination of raw and augmented training data. The trained models are then evaluated by making use of the test data, which is partitioned before processing the data.

**Table 2.** A description of the medical datasets.

| Dataset | Samples | Attribute | Train Set | Test Set |
|---|---|---|---|---|
| Liver | 345 | 7 | 225 | 120 |
| Blood | 748 | 5 | 498 | 250 |
| Haberman | 306 | 4 | 206 | 100 |
| Diabetes | 768 | 8 | 508 | 260 |
| Hepatitis | 155 | 11 | 105 | 50 |
| Breast | 699 | 9 | 459 | 240 |
| Heart | 270 | 14 | 180 | 90 |
| Parkinsons | 195 | 23 | 125 | 70 |
| Phoneme | 5404 | 6 | 3604 | 1800 |
| planningRelax | 182 | 13 | 122 | 60 |
| Saheart | 462 | 9 | 302 | 160 |
| Spectf | 267 | 45 | 177 | 90 |
| WDBC | 569 | 30 | 379 | 190 |

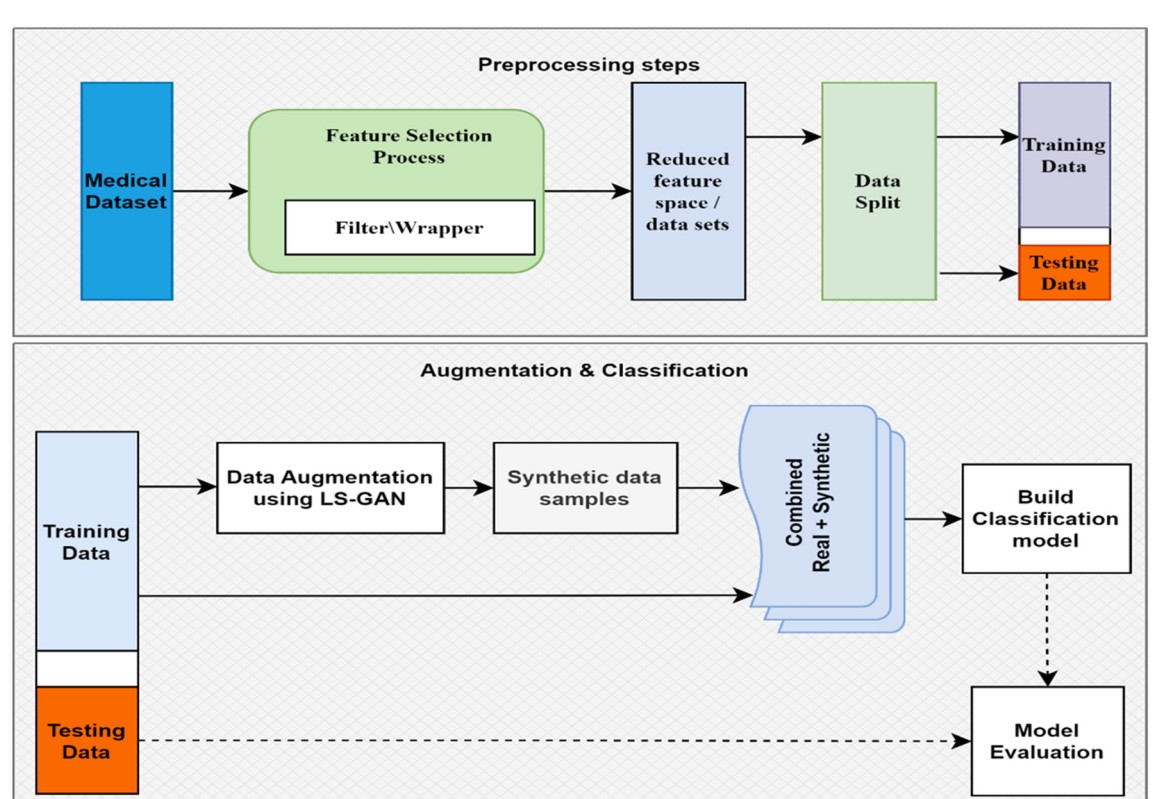

**Figure 1.** Model construction overview.

### 3.3. Feature Selection

In this research, feature selection was used to identify the most relevant features before using the least-square GAN (LS-GAN) to augment the data. The approach first eliminates features that are linearly dependent on other features by using correlation measures to calculate the strength of the linear association between attributes. If the correlation is strong, one of the features is removed. The Boruta feature-selection method is then applied, which uses the random forest algorithm to improve the feature-selection process.

The random forest classifier is constructed by combining multiple decision trees. This process is performed by selecting subsets of the original data and features in each tree. The accuracy of each feature is then evaluated by the outcome of each tree's classification. For the learning process, the least significant features are continually removed during each iteration.

The Boruta algorithm is based on copying each feature and randomly mixing it up (shadow feature) [32]. The original and shadow features are merged before being given to the random forest algorithm. This approach is used to select the most informative features from the datasets and improve the performances of the classification models.

The Boruta algorithm starts by creating a set of "shadow" features, which are copies of the original features with random values. The original and shadow features are then passed to the base classifier (in this research, it was random forest), and their importance scores are calculated. The algorithm then compares the importance scores of the original and shadow features. If the score of a shadow feature is higher than the corresponding original feature, it is considered unimportant and is eliminated. If the score of an original feature is higher than the corresponding shadow feature, it is considered important and is kept. This process is repeated until all the important features are identified. The final set of features is the one that gives the best performance with the base classifier.

The Boruta algorithm is a robust feature-selection method that is particularly useful for datasets with high numbers of irrelevant or redundant features. It is also resistant to overfitting and can handle both continuous and categorical data. The algorithm has been shown to be effective in a variety of applications, such as bioinformatics, genetics, and medical imaging.

### 3.4. Data Augmentation

Sample quality and training stability were the main considerations for using *LS-GAN* [33] in this research. In *LSGAN*s, the discriminator's sigmoid cross-entropy loss function was replaced with the least-square loss function to avoid the vanishing gradient problem. The vanishing gradient problem can occur when utilizing the function of sigmoid cross-entropy loss. Using the function of least-squares loss can move the augmented samples closer to the decision boundary, but these are far from the actual data. The least-square loss function penalizes examples on the right side because of their proximity to the boundary. This model is explained by Equations (1) and (2).

$$Min_{(D)}V_{LSGAN}(D) = \frac{1}{2} E_x \sim_{P_{data}} \left[ D(x) - b^2 \right] + E_{Z \sim P_z^{(z)}} \left[ (D(G(Z)) - A^2 \right] \tag{1}$$

$$Min_{(D)}V_{LSGAN}(G) = \frac{1}{2} E_{Z \sim P_z^{(z)}} \left[ \left( D(G(z)) - c^2 \right) \right] \tag{2}$$

where: $A$ and $b$ represent the labels for fake and actual data, respectively; and $C$ represents the value that $G$ wants $D$ to accept for fake data. The predictive model is better prepared to determine the decision boundaries with enhanced data.

Figure 2 depicts the research model creation procedure. The same procedure is performed for each dataset to generate a model specific to that dataset. The feature-selection process begins by examining all the key features to decide which ones to consider. A filter/wrapper algorithm is applied to produce this result. A subset of the top-ranked features from the dataset is then applied to the GAN augmentation. Finally, the machine learning technique trains and creates the final model using the augmented data.

Figure 2 shows how using more data with a similar real data distribution made the job of the classifier easier. Figure 2a shows that the data belonging to each class were so few that the decision boundary could not be trained enough to determine the right boundary for each class, and in Figure 2b, we can see that using the augmented data allowed the classifier to make more optimum decision boundaries.

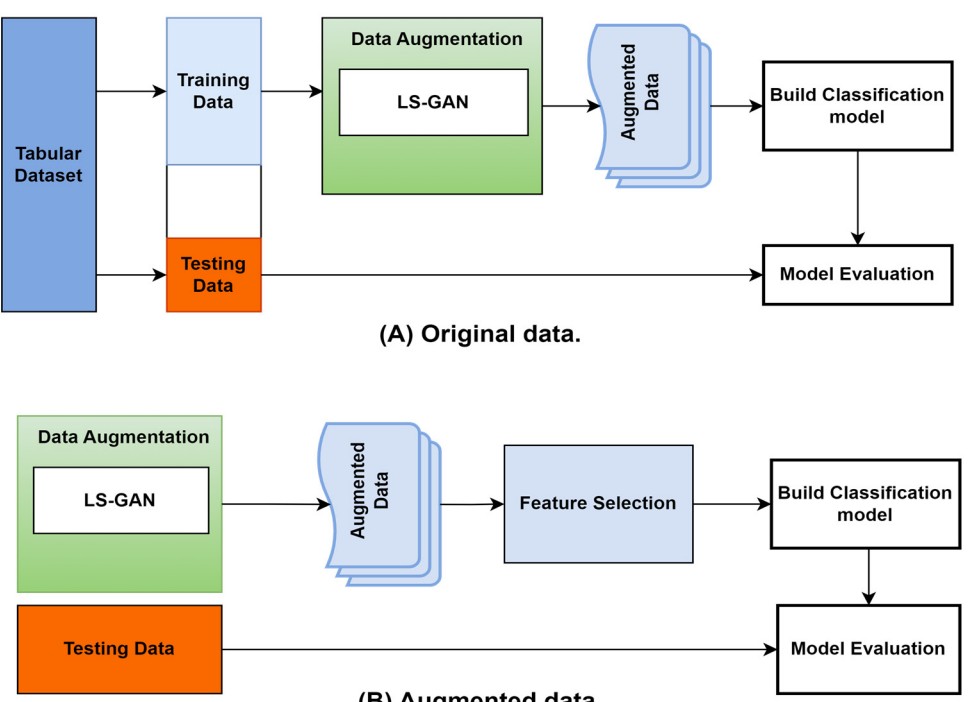

**Figure 2.** Classification boundaries: (**A**) original data and (**B**) augmented data.

Data augmentation generates new data from pre-existing data to increase a training set's size artificially [34]. Data augmentation is used to avoid overfitting, train a model on a small dataset, or get a better model. Data augmentation is used to increase the performances and outcomes of machine learning models by producing additional and diverse instances of the training datasets [34]. The majority of the data augmentation approach was established to work with image datasets. However, not all data-augmentation techniques implemented in the computer vision domain apply to tabular data augmentation. This paper takes advantage of the fact that there is a strong general data-augmentation technique called LS-GAN that is based on a deep network generative model.

Framework-generated data with a similar distribution to actual data and used these data to implement the data-augmentation technique. As a result of training two networks, one for the generator and one for the discriminator, it may generate new samples that are as similar as possible to real ones. Ideally, the discriminator should be unable to tell the difference between augmented and original data. Thus, synthetic data can be generated by generalizing the original data using the augmented samples. This paper considered augmented ratios of [0.1, 1].

The algorithm consists of two main components: the generator and the discriminator. The generator takes a random noise vector as input and generates samples that should resemble the real data distribution. The discriminator takes both real and generated samples and tries to distinguish between them.

The *LSGAN* algorithm can be summarized as follows:

1.  Define the networks: the generator G and the discriminator D.
2.  Initialize the generator and discriminator networks with random weights.
3.  Sample a random noise vector z from a noise distribution $p(z)$.
4.  Generate a fake sample $x' = G(z)$ using the generator network *G*.
5.  Sample a real sample x from the real data distribution.
6.  Pass x and $x'$ through the discriminator network D to get the output values $D(x)$ and $D(x')$.
7.  Calculate the least-squares loss function for the generator and the discriminator:

$$L_D = \frac{1}{2} Ex \sim pdata(x) \left[ (D(x) - 1)^2 \right] + \frac{1}{2} Ez \sim pz(z) \left[ (D(G(z)) - 0)^2 \right] \tag{3}$$

$$L_G = \frac{1}{2} Ez \sim pz(z) \left[ (D(G(z)) - 1)^2 \right] \tag{4}$$

8. Update the generator and discriminator networks using the gradients of the loss functions with respect to the network parameters.
9. Repeat steps 2–7 for a fixed number of iterations or until the desired level of performance is reached.

## 4. Machine Learning Techniques

Machine learning (ML) techniques use an inference principle called induction, which means that general conclusions can be drawn from a small number of examples. Supervised learning is one of the most used ways of induction. A dataset of input and the desired output is used in supervised learning to represent the modelled problem [35]. For each new input, the ML algorithm uses this knowledge representation extracted from the examples to produce the proper output. In this case, when there are $n$ examples in the form $(Xi; Yi)$, $Xi$ stands for input and $Yi$ represents the output. The obtained classifier can be viewed as a function f, which receives an input $X$ and returns an output prediction $Y$. This model also summarizes the training data quickly. The following sections provide an overview of the machine learning techniques employed in this study. Each technique has a different bias or approach, and our selection was based on promising examples from different types of learning.

Random forest (RF) is an ensemble method where each model is made up of decision trees that are trained randomly to reduce the correlation between them. RF is easy to train, makes accurate predictions about output, and gives a reasonable estimate of each feature's importance. Compared to a DT, RF can be more complicated to implement due to the increased number of parameters. In contrast, training several deep trees simultaneously can efficiently use resources [36].

The Naive Bayes (NB) uses Bayes' theorem to assign probabilities to categories. The primary shortcoming of this approach is the necessity for independent predictors. In most real-world instances, the predictors are dependent, which reduces the classifier's performance [37]. It also has short training and prediction times.

Logistic regression (LR) classifiers are statistical models that describe the probability of occurrence of a class by fitting a logistic curve to the dataset [38]. Logistic models, logit models, and maximum-entropy classifiers are other names for LR classifiers. LR models have solved many real-world problems successfully and are widely used in statistics.

The support vector machine (SVM) is a method to discover the optimum hyperplane, or decision boundary, that divides several classes [39]. The mission is to identify the most predictable plane to allow subsequent data points to be categorized with more accuracy. An SVM performs well in scenarios when a distinct margin separates the classes and is effective in high-dimensional spaces. Instead, the dataset size is huge, and SVM needs extensive training time.

A computational system based on the brain's structure, processing technique, and ability to learn is known as an artificial neural network (ANN). For simulating biological neurons, remember they are made up of simple processing units. There are a variety of ways to train an ANN. Weights in an artificial neural network are typically adjusted to approach desired outputs from training data [40]. One of the advantages of ANNs is that they can handle a wide range of functions, including linear and non-linear ones. The requirement for parameter tuning and the difficulty in comprehending the concepts learned by the ANN, which are codified in the weights, are disadvantages.

## 5. Results and Discussion

As stated, each algorithm was assessed using various augmented samples before selecting the optimal performance and providing the number of augmented samples contributing to this performance. Different augmentation ratios are used based on the number of samples in the datasets under the experiment. The ratio of (0.1, 0.2...0.9, 1) was used to select the best model among different model results for each classifier. The optimal augmentation ratio was selected based on the results of the machine learning classification algorithms, and the algorithm that performed the best for each dataset was selected for further analysis. This allowed for a comparison of the different algorithms and the optimal ratio for each algorithm, providing a basis for determining the best augmentation ratio for machine learning classification algorithms in general.

Table 3 shows the size of data for each disease after augmentation. The augmented ratio column represents the factor by which the data were increased. In contrast, the data size after the augmentation column shows the resulting size of the data after the augmentation process was applied. For example, the Blood dataset was increased by a factor of 0.7-fold, resulting in 523 samples, and the Breast dataset was increased by 0.8-fold, resulting in 559 samples. The Phoneme dataset was not augmented, resulting in the same number of samples (5404). The remaining datasets were augmented to varying degrees. The most significant increase was for the Diabetes dataset, which was increased by a factor of 0.9-fold, resulting in 692 samples.

**Table 3.** Size of data for each disease after augmentation.

| Dataset | Augmented Ratio | Size of Data after Augmentation |
|---|---|---|
| Blood | 0.7 | 523 |
| Breast | 0.8 | 559 |
| Diabetes | 0.9 | 692 |
| Haberman | 0.1 | 136 |
| Habitat | 0.1 | 155 |
| Heart | 0.5 | 405 |
| Liver | 0.1 | 345 |
| Parkinsons | 0.5 | 292 |
| Phoneme | 1 | 5404 |
| PlanningRelax | 0.1 | 182 |
| Saheart | 0.1 | 462 |
| Spectf | 0.3 | 400 |
| Wdbc | 0.5 | 854 |

In our study, we use rectified linear units as the activation function for the ten hidden layers in both the generator and the discriminator. The sigmoid function acts as the discriminator's final activation and the hyperbolic tangent function as its generator's final activation. The Adam optimizer trains each model at a variable learning rate. The learning rates of the generators and discriminators were 0.0001 and 0.001, respectively. Mini-batch size was 32, and the training epochs numbered 1000.

In all experimental datasets, Figures 3 and 4 show the relationship between average accuracy and the number of improved samples used by the five approaches. Classifier average classification accuracy fluctuates wildly as the number of boosted samples changes, which is an interesting discovery.

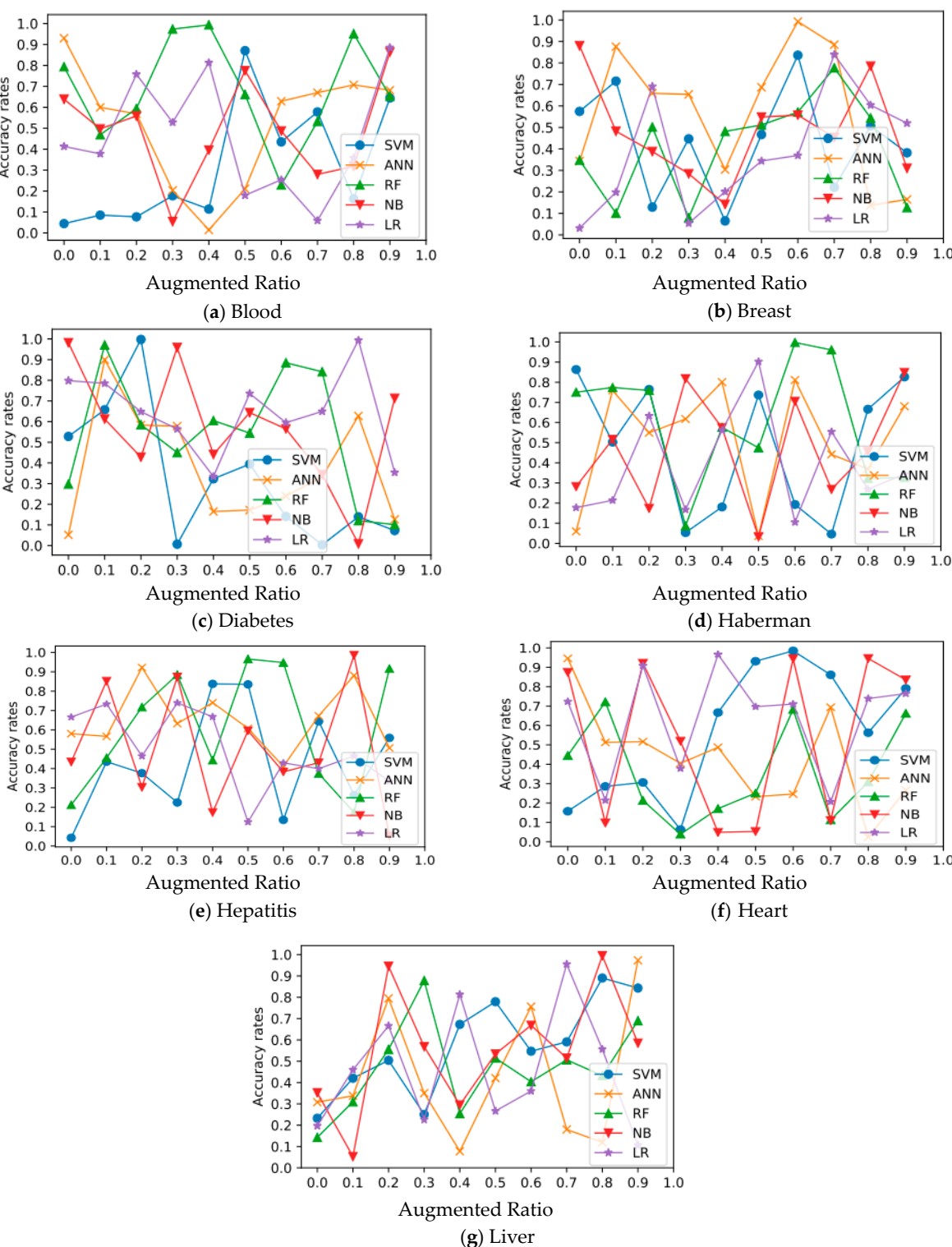

**Figure 3.** Relation of classification and augmentation rates for Diabetes, Blood, Breast, Haberman, Hepatitis, Heart, and Liver datasets.

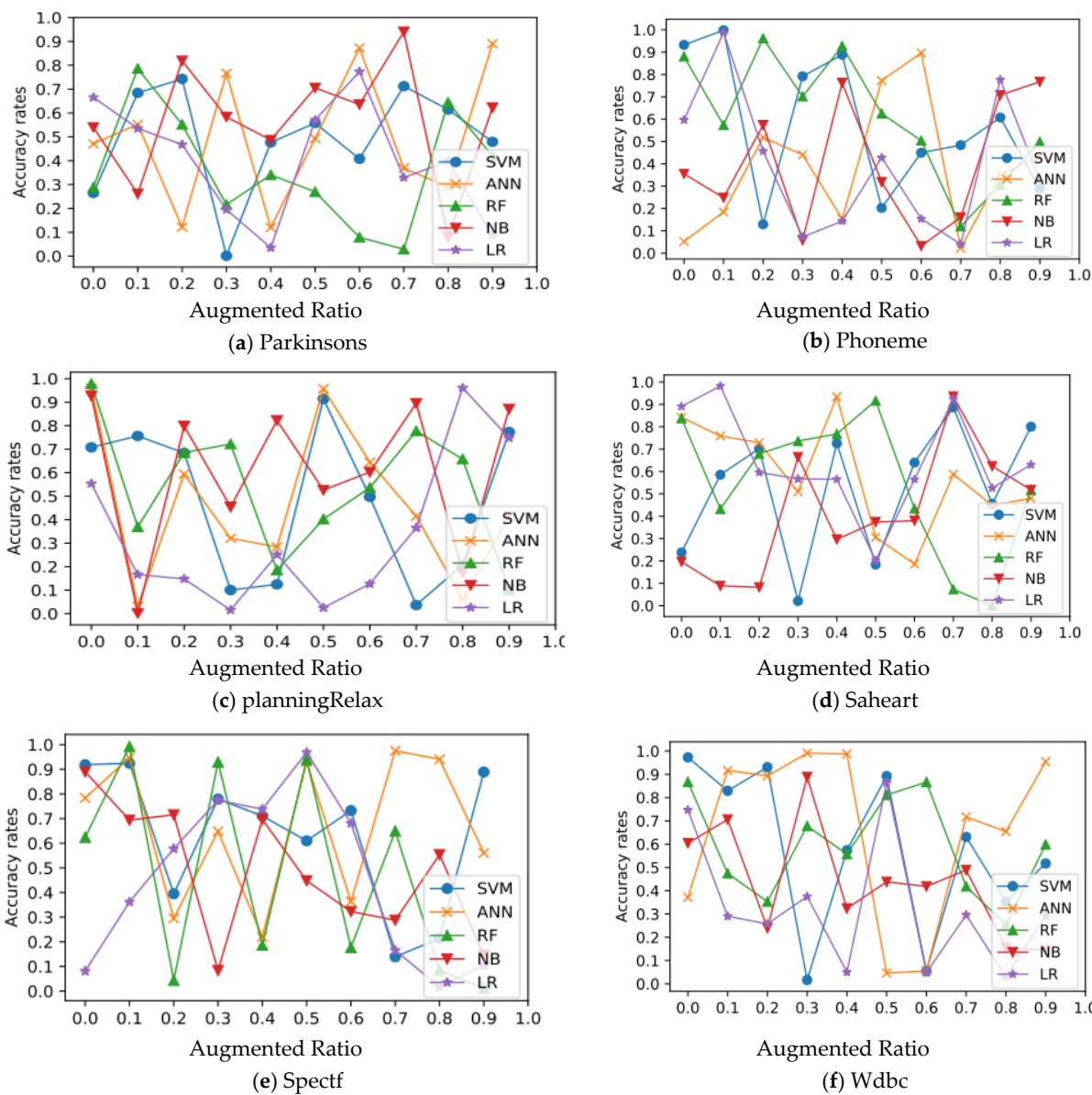

**Figure 4.** Relation of classification and augmentation rates for Parkinson's, Phoneme, planningRelax, Saheart, Spectf, and Wdbc datasets.

The results presented in Table 4 are a summary of the experimental results obtained for different datasets and classifier combinations, including the SVM, the ANN, RF, NB, and LR. The table provides an overview of the average, standard deviation, and best and worst accuracy for each classifier on each dataset. Understanding the best and worst accuracy for different datasets and classifier combinations is essential, as it provides insights into the performances of the classifiers in different scenarios. This information can be useful for choosing the best classifier for a specific task and understanding the strengths and limitations of different classifiers. Additionally, the augmented ratio can also give a rough estimate of the robustness of each classifier.

**Table 4.** Summary of the experimental results obtained for the SVM, the ANN, the RF, NB, and LR for all datasets.

| Dataset | Measure | SVM | ANN | RF | NB | LR |
|---------|---------|-----|-----|-----|-----|-----|
| Blood | Avg | 0.7978 | 0.7844 | 0.8035 | 0.8005 | 0.7982 |
| | Stdv | 0.0075 | 0.0029 | 0.0057 | 0.0087 | 0.008 |
| | Best | 0.8089 | 0.7854 | 0.8129 | 0.8168 | 0.8089 |
| | Worse | 0.7815 | 0.7815 | 0.7893 | 0.7815 | 0.7854 |
| | Augmented Ratio | 0.7 | 1 | 0.2 | 0.2 | 0.6 |
| Breast | Avg | 0.9738 | 0.9627 | 0.9741 | 0.9629 | 0.9757 |
| | Stdv | 0.0056 | 0.0078 | 0.0075 | 0.0128 | 0.006 |
| | Best | 0.9801 | 0.9675 | 0.9843 | 0.9885 | 0.9843 |
| | Worse | 0.9591 | 0.9465 | 0.9633 | 0.9423 | 0.9633 |
| | Augmented Ratio | 0.8 | 0.5 | 0.5 | 0.8 | 0.8 |
| Diabetes | Avg | 0.7726 | 0.7258 | 0.7659 | 0.7554 | 0.7636 |
| | Stdv | 0.0128 | 0.0088 | 0.013 | 0.0203 | 0.0148 |
| | Best | 0.7824 | 0.7377 | 0.7928 | 0.7876 | 0.798 |
| | Worse | 0.7407 | 0.6996 | 0.7407 | 0.7094 | 0.7355 |
| | Augmented Ratio | 0.9 | 1 | 0.8 | 0.6 | 0.7 |
| Haberman | Avg | 0.7313 | 0.7306 | 0.7319 | 0.7281 | 0.73 |
| | Stdv | 0.0057 | 0.007 | 0.0101 | 0.0208 | 0.0093 |
| | Best | 0.7344 | 0.7344 | 0.7535 | 0.763 | 0.744 |
| | Worse | 0.7249 | 0.7154 | 0.7154 | 0.6963 | 0.7154 |
| | Augmented Ratio | 0.1 | 0.9 | 0.1 | 0.3 | 0.2 |
| Hepatitis | Avg | 0.9124 | 0.8759 | 0.8747 | 0.8489 | 0.847 |
| | Stdv | 0.0204 | 0.0193 | 0.0279 | 0.035 | 0.0329 |
| | Best | 0.9256 | 0.9256 | 0.9256 | 0.9256 | 0.9445 |
| | Worse | 0.869 | 0.8502 | 0.8313 | 0.7936 | 0.8124 |
| | Augmented Ratio | 0.1 | 0.4 | 0.1 | 0.1 | 0.2 |
| Heart | Avg | 0.8623 | 0.8413 | 0.7978 | 0.7754 | 0.8478 |
| | Stdv | 0.033 | 0.0348 | 0.0335 | 0.0414 | 0.0183 |
| | Best | 0.8924 | 0.8707 | 0.8598 | 0.8598 | 0.8707 |
| | Worse | 0.8163 | 0.7837 | 0.7185 | 0.6859 | 0.8055 |
| | Augmented Ratio | 0.5 | 0.5 | 0.2 | 0.4 | 0.3 |
| Liver | Avg | 0.7333 | 0.6921 | 0.7014 | 0.6972 | 0.7203 |
| | Stdv | 0.0369 | 0.0356 | 0.038 | 0.0513 | 0.0273 |
| | Best | 0.7892 | 0.713 | 0.7808 | 0.7638 | 0.7723 |
| | Worse | 0.6197 | 0.5859 | 0.6113 | 0.5689 | 0.6621 |
| | Augmented Ratio | 0.1 | 0.2 | 0.5 | 0.1 | 0.3 |
| Parkinsons | Avg | 0.9071 | 0.8902 | 0.8882 | 0.8543 | 0.8717 |
| | Stdv | 0.013 | 0.0264 | 0.0242 | 0.0375 | 0.0263 |
| | Best | 0.9265 | 0.9116 | 0.9265 | 0.8966 | 0.9116 |
| | Worse | 0.8668 | 0.8519 | 0.8071 | 0.7474 | 0.822 |
| | Augmented Ratio | 0.5 | 0.7 | 0.6 | 0.7 | 0.5 |

**Table 4.** *Cont.*

| Dataset | Measure | SVM | ANN | RF | NB | LR |
|---|---|---|---|---|---|---|
| Phoneme | Avg | 0.8357 | 0.8325 | 0.836 | 0.8387 | 0.8243 |
| | Stdv | 0.007 | 0.0091 | 0.0074 | 0.0078 | 0.0088 |
| | Best | 0.846 | 0.839 | 0.8504 | 0.8482 | 0.8373 |
| | Worse | 0.8205 | 0.8118 | 0.8232 | 0.8259 | 0.7993 |
| | Augmented Ratio | 1 | 1 | 1 | 1 | 1 |
| PlanningRelax | Avg | 0.6565 | 0.6549 | 0.6484 | 0.6221 | 0.6554 |
| | Stdv | 0.009 | 0.0093 | 0.0179 | 0.0354 | 0.0121 |
| | Best | 0.6624 | 0.6624 | 0.6785 | 0.6785 | 0.6785 |
| | Worse | 0.6463 | 0.6463 | 0.614 | 0.5334 | 0.6301 |
| | Augmented Ratio | 0.1 | 0.7 | 0.1 | 0.2 | 0.2 |
| Saheart | Avg | 0.7541 | 0.7553 | 0.7342 | 0.7332 | 0.7484 |
| | Stdv | 0.0115 | 0.0099 | 0.0228 | 0.0235 | 0.0161 |
| | Best | 0.7669 | 0.7606 | 0.7733 | 0.7733 | 0.7796 |
| | Worse | 0.729 | 0.7353 | 0.6973 | 0.6846 | 0.7226 |
| | Augmented Ratio | 0.1 | 0.8 | 0.1 | 0.1 | 0.8 |
| Spectf | Avg | 0.7835 | 0.7872 | 0.7777 | 0.767 | 0.7828 |
| | Stdv | 0.0173 | 0.0067 | 0.0201 | 0.0244 | 0.0067 |
| | Best | 0.7923 | 0.7923 | 0.8033 | 0.8143 | 0.7923 |
| | Worse | 0.7374 | 0.7813 | 0.7264 | 0.7154 | 0.7703 |
| | Augmented Ratio | 0.3 | 0.4 | 0.1 | 1 | 0.3 |
| Wdbc | Avg | 0.9624 | 0.9513 | 0.9549 | 0.9528 | 0.9588 |
| | Stdv | 0.0059 | 0.0123 | 0.0117 | 0.0125 | 0.0113 |
| | Best | 0.9753 | 0.9599 | 0.9856 | 0.9753 | 0.9753 |
| | Worse | 0.9547 | 0.9289 | 0.9341 | 0.9186 | 0.9392 |
| | Augmented Ratio | 0.5 | 0.7 | 0.4 | 0.5 | 0.5 |

According to the research findings, adding augmentation samples reduced classification precision on the datasets. In contrast, the RF surpassed the average performances of the other training models. The SVM, RF, ANN, NB, and LR algorithms were evaluated using 13 datasets in Table 4. The tables show how many augmentation rates are required to achieve a certain classification accuracy level. When comparing the average classification accuracies of all algorithms, we can see that the SVM outperformed the competition on 9 of 13 datasets.

Furthermore, SVM outperformed other methods on various datasets, including Habitat, liver, and Parkinson's. According to the findings, only two datasets, Haberman and Blood, showed that the ANN was the superior algorithm. In contrast, only one dataset, Phoneme, showed that NB was superior to the other algorithms. Additionally, the SVM, ANN, and NB algorithms failed to surpass LR in one scenario (Breast).

In addition, the SVM surpassed the RF on nearly all datasets tested. Only on two datasets did the RF outperform the SVM (Saheart and Spectf). In addition, the SVM had improved accuracy and used fewer augmented samples in most situations. As a result, SVM outperformed all other classification algorithms in 60% of cases in the planningRelax, Habitat, Liver, Parkinsons, Phoneme, WDBC, and Saheart datasets. While SVM produced an average accuracy of roughly 91.13 on the Habitat dataset, the same model achieved an average of 84.6% on the same dataset with just a 0.01 enhancement ratio.

Regarding the Liver dataset, the SVM achieved 73.3 accuracy with only a 0.01 enhancement ratios, whereas LR reached 71.9 accuracy with a 0.03 enhancement. The classification accuracy of the SVM was higher on 13 datasets than that of LS, according to the comparison. The SVM came second on the Blood and Breast datasets, but only by a narrow margin (LS won). The SVM used a nearly constantly increased ratio in 10 of the 13 datasets. There were 11 datasets on which the SVM outperformed the ANN, whereas there were four on which the SVM beat the ANN.

In 9 of the 11 datasets, the SVM fared best with lesser or equal augmentation ratios. Although the SVM beat NB on 13 datasets, only on the Blood and Phoneme datasets did NB outperform the SVM. Overall, the SVM needed more samples in all 12 datasets than NN.

Boxplots are used in this work to showcase the performances of the machine learning algorithms (SVM, RF, ANN, NB, and LR) used for the classification of medical datasets. Figures 5 and 6 present the box plots that demonstrate the variation in each algorithm's accuracy over all the datasets. The boxplots consist of boxes representing each of the 20 tests that were conducted differently. Each box represents the distribution of the accuracy rate of each algorithm. The top and bottom of each box represent the 75th and 25th percentiles, respectively, and the line in the middle represents the median accuracy. The boxplots clearly showcase the stability and superiority of the SVM compared to the other algorithms, making it evident that the SVM is the best algorithm for the classification of medical datasets in this work.

The given experiment results show the performances of different machine learning models (SVM, ANN, RF, NB, and LR) on various datasets (Blood, Breast, Diabetes, Haberman, Habitat, Heart, Liver, and Parkinsons). The performance of each model was measured in terms of the average, standard deviation, best and worst accuracy, and the augmented ratio (ratio of augmented data to original data). In general, it appears that the random forest (RF) and the support vector machine (SVM) models had the highest average accuracy considering all datasets. The RF had the highest average accuracy for the Blood, Breast, Diabetes, Haberman, Habitat, and Liver datasets; and SVM had the highest average accuracy for the Heart dataset. The logistic regression (LR) model had the lowest average accuracy on the datasets.

The standard deviations of the accuracy for the RF and SVM models are relatively low, indicating that these models had consistent performances. On the other hand, the standard deviations of the accuracy for the ANN and LR models are relatively high, indicating that these models had less consistent performances.

The best accuracies for the RF model were obtained on the Breast, Diabetes, Haberman, and Liver datasets. The best accuracy for the SVM model was obtained on the Heart dataset. The best accuracy for the ANN model was obtained on the Breast dataset. The best accuracy for the NB model was obtained on the Diabetes dataset. Additionally, the best accuracy for the LR model was obtained on the Heart dataset. The worst accuracy for the RF, SVM, ANN, and NB models was obtained on the Liver dataset. The worst accuracy for the LR model was obtained on the Habitat dataset.

From the results, it can be observed that the performances of the models varied on the different datasets. On average, the random forest (RF) model had the highest performance on the datasets, followed by the support vector machine (SVM) model. The Naive Bayes (NB) model had the lowest performance on average. However, it is important to note that the performances of the models also varied within each dataset.

In terms of the augmented ratio, it appears that the datasets were augmented differently. Some datasets had higher ratios of augmented data to original data than others. The datasets that had the highest augmented ratios were the Diabetes, Habitat, and Liver datasets, which all had a ratio of 0.8 or higher. The datasets that had the lowest augmented ratio were the Blood, Breast, and Haberman datasets, which all had a ratio of 0.5 or lower. The results suggest that the RF and SVM models have the best overall performance in terms of average accuracy and consistency of performance on the datasets tested. However, it is

important to note that the results may have been affected by the specific characteristics of the datasets and the way they were augmented.

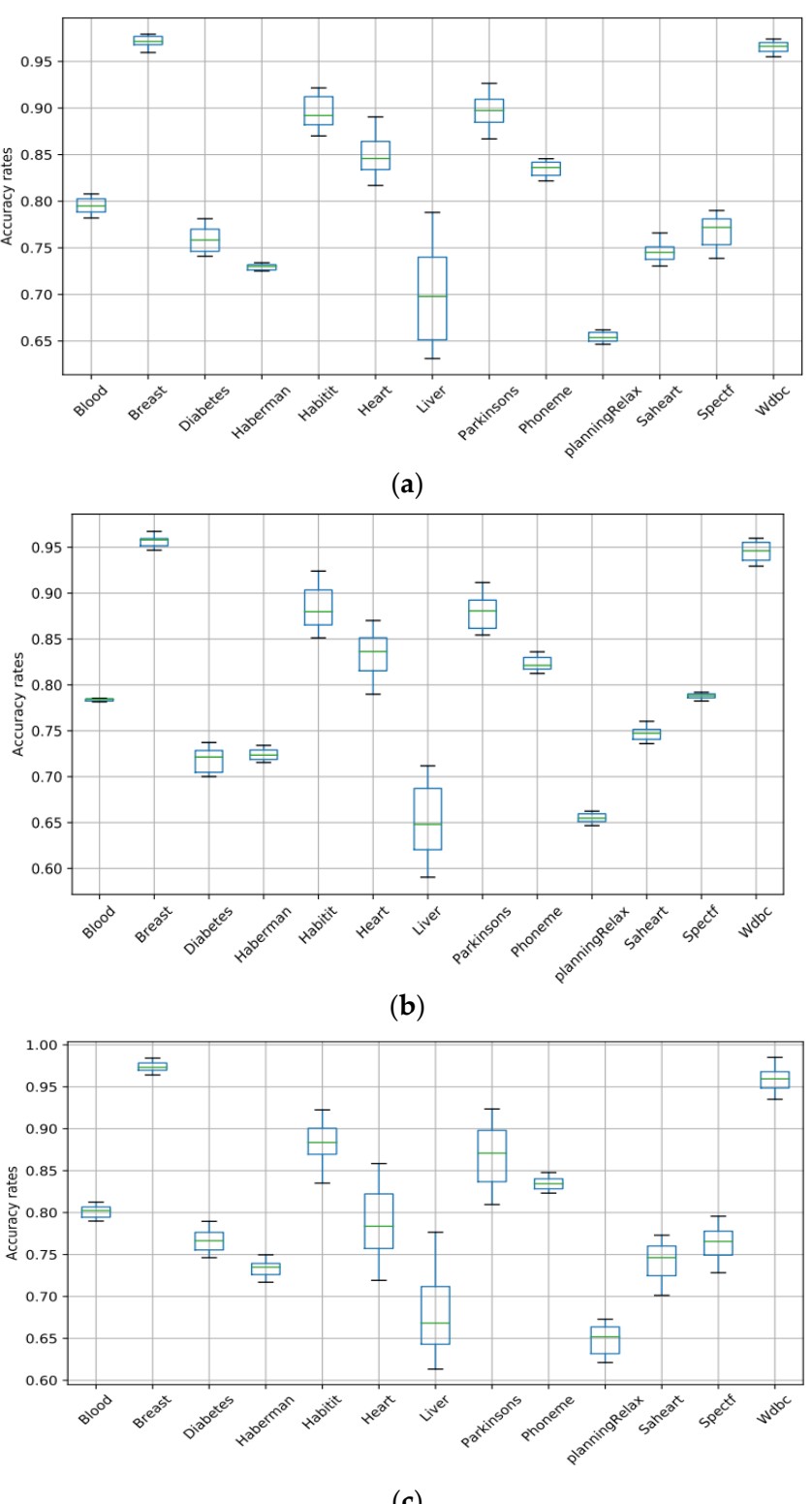

**Figure 5.** Box plots of the SVM, RF, and ANN for all datasets. (**a**) SVM; (**b**) RF; (**c**) ANN.

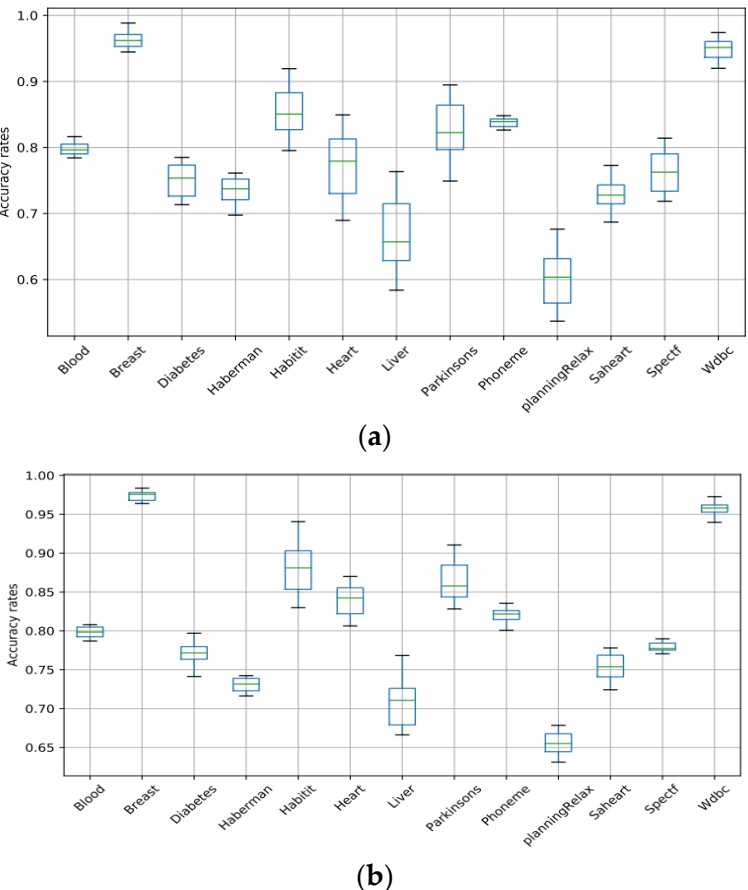

**Figure 6.** Box plots of NB and LR for all datasets. (**a**) NB; (**b**) LR.

The performances of different classifiers in this study were evaluated using several metrics, including average accuracy, standard deviation, best accuracy, and worst accuracy. The results show that the SVM model outperformed all other classifiers used in the study. This superior performance of the SVM model can be attributed to its ability to handle non-linearly separable data by transforming them into a higher dimensional space where they can be separated linearly. This transformation allows the SVM model to capture complex patterns in the data, which is particularly useful in the medical field, where data can be highly non-linear. Additionally, the SVM model also has a regularization parameter that helps avoid overfitting, a common issue in small medical datasets. The regularization parameter controls the trade-off between fitting the model to the training data and avoiding overfitting. The SVM model also has a kernel function that allows it to handle non-linear relationships in the data.

The Wilcoxon signed-rank test was used to see if there was a significant difference between the SVM, and the RF and other methods. Wilcoxon is a nonparametric paired test more resistant to outliers than the *t*-test [41]. There were no significant differences between the SVM and the other classifiers in *p*-values (Table 5). In addition, Table 5 demonstrates that the SVM performed superiorly to the ANN on eight different datasets, including Heart, WDBC, Diabetes, Parkinson's, HepatitisLiver, and planningRelax. Based on test outcomes, it looks as though the ANN did better on the Blood dataset. While SVM outperformed NB in all ten datasets tested (Breast, Diabetes, and Habitat), NB could not beat the SVM on any of the other datasets (parkinson's disease, planningRelax, heart, Spectf, and WDBC). In addition, Table 5 shows that the SVM performed better than the RF on eight datasets. Finally, the SVM outperformed LR on the following five datasets: Diabetes, Habitat, Parkinson's, Phoneme, and Spectf. In contrast, LR did not perform better than the SVM in any tested

datasets. According to the comprehensive comparison analysis, the SVM is superior to NB and the ANN in classification accuracy and the number of enhanced samples needed.

**Table 5.** The Wilcoxon test's *p*-values for the SVM's accuracy results compared to those of other methods.

| Dataset | SVM-ANN | SVM-NB | SVM-RF | SVM-LS |
|---|---|---|---|---|
| Blood | 0.0016 | 0.1023 | $1.21 \times 10^9$ | 0.8088 |
| Breast | 0.9506 | 0.0095 | $1.90 \times 10^9$ | 0.0949 |
| Diabetes | 0.0030 | 0.0015 | $1.77 \times 10^{11}$ | 0.0024 |
| Hepatitis | 0.001 | 0.0 | $3.18 \times 10^8$ | 0 |
| Heart | 0.0 | 0.0 | 0.010933 | 0.1205 |
| Liver | 0.0132 | 0.0101 | $5.18 \times 10^7$ | 0.1316 |
| Parkinsons | 0.0015 | 0.0 | 0.0043 | 0.0 |
| Phoneme | 0.930 | 0.0933 | 0.1184 | 0.0 |
| planningRelax | 0.0 | 0 | 0.4411 | 0.6013 |
| Saheart | 0.0027 | 0.0054 | 0.79501 | 0.0506 |
| Wdbc | 0.0042 | 0.0037 | $5.28 \times 10^6$ | 0.1495 |
| Spectf | 0.1060 | 0.0060 | 0.6535 | 0.018 |

In conclusion, based on the results of the Wilcoxon signed-rank test presented in Table 5, it can be seen that the SVM classifier performs significantly better than the other classifiers on most of the datasets tested. The *p*-values for the SVM's accuracy compared to the accuracy of other methods, such as the ANN, NB, the RF, and LS, indicate statistically significant differences in performance. Specifically, the *p*-values for the SVM compared to the ANN, NB, and LS are all less than 0.05, indicating that the SVM is significantly better than these classifiers on all datasets. In addition, the *p*-values for the SVM compared to the RF are mostly less than 0.05, indicating that the SVM is significantly better than RF on most of the datasets. It can be seen that the SVM is superior to the ANN, NB, and LR in classification accuracy and the number of enhanced samples needed. In addition, the SVM was also found to be better than the RF on most of the datasets. Therefore, it can be concluded that the SVM is a better classifier than the other classifiers on most of the datasets.

The least-square GAN (LS-GAN) algorithm has potential benefits for data augmentation of small medical datasets, but it also has some limitations. One of the main benefits of using LS-GANs is that it can generate synthetic samples that closely resemble the real data, improving the classification model's generalization performance. Additionally, LS-GANs can stabilize the classifier and provide a more precise indication of the number of samples needed by the model, thereby avoiding overfitting due to the small number of the labeled samples. However, some limitations exist to using LS-GANs in small medical datasets. One major limitation is that LS-GANs can be difficult to train due to the sensitivity of the least-squares loss function to the choices of hyperparameters. Additionally, GANs can generate samples that are not realistic; this is called mode collapse, which means that the generator produces limited variations of the same image. This results in a biased model that cannot generalize well to unseen images.

In general, minimizing the objective function of a standard GAN suffers from the vanishing gradients in GANs models, making it difficult to update the generator. This problem can be solved by LS-GANs, which penalize samples based on how close they are to the decision boundary. This makes more gradients that can be used to update the generator.

The time complexity of the LS-GAN can be described as $O(n)$, where $n$ is the number of iterations used in training the model. The space complexity of LS-GAN is $O(m)$, where m is the size of the input data. The LS-GAN algorithm requires significant computational resources, especially in terms of memory usage, as it requires storing intermediate values during training.

Furthermore, GANs are computationally expensive, which can be a limitation when working with large datasets. Additionally, GANs are sensitive to their initialization, and their generator can converge to a suboptimal solution. Therefore, it is important to carefully evaluate the performance of a GAN-augmented model and consider the trade-off between the benefits and limitations of using LS-GANs in small medical datasets.

## 6. Conclusions

The paper introduced an augmentation framework based on a generative adversarial network (GAN). The proposed framework was thoroughly evaluated on 13 medical datasets, and the results showed that the GAN-based data augmentation strategy could significantly enhance the performances of all classifiers. Specifically, the experiments revealed that the support vector machine (SVM) outperformed other methods by accurately classifying data in most cases while using fewer augmented samples. Moreover, the SVM proved more resilient and stable than other methods. These findings demonstrate that the proposed system can effectively and reliably produce tabular medical data and train machine learning-based classifiers for classification applications. While the results of this study are promising, there is still room for improvement. Future research could explore alternative loss functions that may be less sensitive to hyperparameters and more robust to mode collapse. Additionally, researchers could investigate ways to make GANs more efficient and less computationally expensive.

**Author Contributions:** Conceptualization, M.A. and A.A.-q.; methodology, M.A. and A.A. (Ayoub Alsarhan); software, M.A., B.S., A.A.-q., A.A. (Ayoub Alsarhan), A.A. (Amjad Aldweesh) and M.E.; validation, M.A., M.E., A.A.-q., A.A. (Ayoub Alsarhan), A.A. (Amjad Aldweesh) and N.A.; formal analysis, M.A.; investigation, M.A.; resources, M.A. and A.A. (Amjad Aldweesh); data curation, M.A.; writing—original draft preparation, M.A., A.A.-q., A.A. (Ayoub Alsarhan), A.A. (Amjad Aldweesh), M.E. and B.S.; writing—review and editing, M.A.; visualization, M.A.; supervision, A.A.-q., A.A. (Ayoub Alsarhan), A.A. (Amjad Aldweesh) and N.A.; project administration, M.A.; funding acquisition, A.A.-q., A.A. (Ayoub Alsarhan), A.A. (Amjad Aldweesh). All authors have read and agreed to the published version of the manuscript.

**Funding:** This research received no external funding.

**Institutional Review Board Statement:** Not applicable.

**Informed Consent Statement:** Not applicable.

**Data Availability Statement:** Not applicable.

**Acknowledgments:** Amjad Aldweesh would like to thank the Deanship of Scientific Research at Shaqra University (KSA) for supporting this research and Mohammad Alauthman would like to thank The University of Petra (Jordan) for supporting this research.

**Conflicts of Interest:** The authors declare they have no conflict of interest to report regarding the present study.

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
