# Peer review of "Enhancing Small Medical Dataset Classification Performance Using GAN"

_informatics, doi:10.3390/informatics10010028_

Round 1

Reviewer 1 Report

The paper provides a comprehensive overview of the current state-of-the-art medical data classification and augmentation techniques and demonstrates how the proposed GAN-based data augmentation framework can enhance the performance of small medical dataset classification. The Least-Square GAN (LS-GAN) choice for data augmentation is well-motivated, and its performance on 13 different medical datasets is thoroughly evaluated. 

Overall, I believe the paper represents a significant contribution to the field of medical dataset classification and has the potential to inspire further research in this area. I do have a few minor comments that would be improve the paper quality and presentation:

1. Clarify the motivation and the significance of the study in the introduction section.

2. It would be helpful to provide more detailed information about the 13 medical datasets used in the experiments, including their sizes and the number of classes.

3. It would be interesting to compare the results of the proposed framework with the results of other augmentation techniques, such as synthetic oversampling or random transformations.

4. The discussion of the limitations of the proposed framework and possible future directions could be further expanded.

5. Check the paper for typos, grammatical errors, and clarity issues.

6. It is interesting to add information about the computational complexity of the proposed solution.

Author Response

Thank you very much for your review and feedback on our paper. We are pleased to hear that you found our work to be a significant contribution to the field of medical dataset classification and that our GAN-based data augmentation framework is well-motivated and thoroughly evaluated.

Reviewer 2 Report

1.     Abstract should be precise and should clearly show the background, hypothesis, problem statements, methodology, techniques used with the relevant method, and the outcome.

2.     There has to be mention of clear motivation and objectives for this work

3.     Caption of Figure 3 shows (a) and (b), which are not shown in the figure itself.

4.     Size of data for each disease after augmentation should be mentioned in the paper

5.     Table 3 is not clear, it shows the average, stdv, best and worst. What does it represent?

6.     How augmentation ratio is calculated?

7.     It is important to show some examples of datasets used.

8.     Is figure 1 self-created?

9.     It would be good to extend the analysis of the results with an explanation of the observed behavior. For eg Why SVM is better than all other classifiers used in the work?

10.  The contributions of this work need to be clearly articulated. The author should consider justifying the performance of this study with recent studies and methods.

11.  It should be clearly mentioned that how boxplots are used in the work to showcase the model performances.

12.  It is advised to cite the latest research work (2021, 2022, 2023), most of the references are of previous years.

13.  I would suggest adding a new discussion to show the advantage. The following studies can be considered

a.     A Bottom-Up Review of Image Analysis Methods for Suspicious Region Detection in Mammograms

b.     Deep convolutional neural networks for computer-aided breast cancer diagnostic: a survey

c.     Image Augmentation Techniques for Mammogram Analysis

d.     Computer-Aided Breast Cancer Diagnosis: A Comparative Analysis of Breast Imaging Modalities and Mammogram Repositories.

e.     Deep ensemble transfer learning-based framework for mammographic image classification

Author Response

Thank you very much for your review and feedback on our paper. 

Reviewer 3 Report

Overall the work is novel and interesting.

A few comments are as follows:

1. The reason for best and least accuracy for different datasets and classifier combination should be explained

2. The number of false negative and false positive images that are augmented or the number of images that are augmented in each positive and negative category can be listed

3. For each dataset, couple of images and their different augmented images can be shown. Basically no images from the dataset are shown in the paper. 

Author Response

Thank you very much for your review and feedback on our paper. Our response to your valuable comments in the attached. 

Reviewer 4 Report

Dear Authors,

All comments are in the attached file. The paper can be accepted with minor changes.

​Kind Regards, Dr Sophiya Rumovskaya

Author Response

Thank you very much for your review and feedback on our paper. Kindly, refer to the attached which addresses your valuable comments. 

Round 2

Reviewer 2 Report

1.     The manuscript is revised as per the comments given, but there are still some key issues.

2.     Authors should check whether figure 1 can be used freely or permission from the source is required.

3.     Since the authors are saying the proposed work is a “novel approach for data augmentation in small medical datasets using Generative Adversarial Networks” The author should consider justifying the performance of this study with recent similar studies and methods.

4.     I would suggest adding a new discussion to show the advantage. The following studies can be considered

a.     A Bottom-Up Review of Image Analysis Methods for Suspicious Region Detection in Mammograms

b.     Deep convolutional neural networks for computer-aided breast cancer diagnostic: a survey

c.     Image Augmentation Techniques for Mammogram Analysis

Author Response

We would like to thank the reviewer for their valuable comments and please refer to the attached reply. 
